# A New Approach to the Identification of Palliative Care Needs and Advanced Chronic Patients among Nursing Home Residents

**DOI:** 10.3390/ijerph18063171

**Published:** 2021-03-19

**Authors:** Ana A. Esteban-Burgos, María José Lozano-Terrón, Daniel Puente-Fernandez, César Hueso-Montoro, Rafael Montoya-Juárez, María P. García-Caro

**Affiliations:** 1Department of Nursing, University of Granada, 18016 Granada, Spain; anaestebanburgos@ugr.es (A.A.E.-B.); cesarhueso@ugr.es (C.H.-M.); rmontoya@ugr.es (R.M.-J.); mpazgc@ugr.es (M.P.G.-C.); 2Doctoral Programme in Clinical Medicine and Public Health, University of Granada, 18012 Granada, Spain; 3Caxar de la Vega Nursing Home, Cajar, 18199 Granada, Spain; psicologa1@caxardelavega.es; 4Mind, Brain and Behaviour Research Institute, University of Granada, 18071 Granada, Spain

**Keywords:** palliative care, chronic patient, nursing homes, palliative needs, elderly, residents, advanced chronicity, frailty, complexity, prognosis

## Abstract

Background: Proper planning of Palliative Care in nursing homes requires advanced knowledge of the care needs that residents show. The aim of the study was to evaluate Palliative Needs and other conditions such as fragility, complexity, and prognosis and also to suggest new indicators for the establishment of the resident’s advanced chronic condition. Methods: Cross-sectional study conducted in 149 nursing homes Complex Chronic residents evaluated by trained professionals. Palliative Care Needs, assessed by the NECPAL ICO-CCOMS© tool, and fragility, case and palliative complexity and prognosis were evaluate through a comprehensive assessment. Descriptive analyses and association measures were performed setting the statistical significance at 0.05. Results: More than 50% of the residents had positive Surprise Question and other Palliative Needs and were classified as Advanced Chronic Patients. Distress and/or Severe Adaptative Disorder was the most frequent need shown by the residents and significant differences in levels of frailty and other characteristics, were found between the Positive and the Negative Surprise Question Groups. Statistically significant correlations were also found between aspects of both groups. Conclusions: Nursing homes residents show Palliative Needs regardless of the response to the Surprise Question of the NECPAL tool. Other characteristics such as presence of an intermediate level of frailty are suggested as a new perspective to identify advanced chronic patients among nursing homes residents.

## 1. Introduction

Changes in the ageing of the population and, by extension, in patterns of disease have increased the relevance of nursing homes as places where the elderly are treated and cared for. Currently, the majority of deaths in this population group are caused by cancer and chronic diseases, combined in many cases with concomitant diseases [1].

In high-income countries, more than half of older individuals have multimorbidity, with prevalence markedly increasing in very old age [2]. According to the Spanish National Institute of Statistics (2020) [3], life expectancy at birth in Spain stood at 86.3 years for women and 80.7 years for men in 2018. The population aged 65 and above is projected to reach 26.5% of the total population by 2035, 8.1% of whom will be aged 80 and above [4].

Both the prevalence and mortality rate of individuals with advanced chronic disease are increasing exponentially and are frequently associated with conditions such as dependency, frailty, and multimorbidity, with varying degrees of complexity in terms of needs and demands. Therefore, a significant increase in healthcare and long-term care needs among the elderly is to be expected. Multiple studies [5,6] have reported that, in recent years, nursing homes have become the place where many older individuals will spend their final days.

Proper planning of palliative care in nursing homes requires advance clarification of the treatment objectives for residents needing this care [7]. A palliative approach is considered suitable when the resident’s condition is incurable and the symptoms of the disease call for effective management. The aim is to improve the resident’s comfort and performance and meet their psychological, spiritual, and social needs.

In general terms, a complex chronic patient has been described as an individual whose clinical management is perceived as particularly difficult by their healthcare providers due to characteristics relating to the patients themselves (morbidity profile, progression of the condition, resource consumption, etc.), to professionals (frequent transitions, conflicting care criteria, situations of clinical uncertainty), and to the people around the patient (adverse psychosocial circumstances). An advanced chronic patient (ACP) may be defined as a patient whose complexity is associated with limited life expectancy and increasing palliative care needs (PCNs) [8,9].

In order to facilitate care planning in accordance with the complexity of the needs to be met and the resources required, two transition moments have been identified [9]:Transition from complex chronicity to advanced chronicity: The main triggers of this transition are limited life expectancy and increasing PCNs or demands, such as symptom control, futile care management, adequacy of therapeutic efforts, identification of values and preferences, spiritual care, etc.Transition from advanced chronicity to terminal illness: Unequivocally limited life expectancy, predominance of palliative care, extreme communication and support needs, management of emotional and spiritual needs, management of grief and practical decisions regarding impending death, and emotional support for families.

At present, there are no accurate statistics on the exact number of residents in need of palliative care, their level of complexity, or the use and adequacy of the services they are provided with [10,11]. However, the prevalence of chronically ill patients with PCNs has been estimated at 1.4% of the general population, 26–40% of whom are in acute care hospitals and 60–70% in health and social care centers [12].

PCNs can be met through the implementation of end-of-life care programs in nursing homes, as evidenced by established European programs such as the one implemented as part of the PACE study (Palliative Care for Older People) [13,14].

The NU-HELP program (Nursing Homes End-of-Life Program) (AP-0105-2016), which provides the framework for this study, aims to evaluate the effectiveness of the implementation of an ad hoc end-of-life care program in nursing homes in improving the quality of end-of-life care in nursing homes in Andalusia, Spain.

Before incorporating residents into such programs, an initial assessment is necessary to obtain in-depth knowledge of their clinical and psychosocial needs. This provides professionals with all the information they need to deliver care tailored to residents’ needs.

In this study, we propose a comprehensive assessment of residents based on sociodemographic variables and instruments and indices that have proven helpful in determining the needs of patients who are eligible for inclusion in end-of-life care programs in nursing homes.

This proposal revolves around the assessment of PCNs using the NECPAL-ICO-CCOMS©3.1 instrument [11], which has been shown to be helpful in establishing the advanced chronic status of patients by identifying PCNs and limited life expectancy in a variety of clinical settings, including nursing homes [11,15]. The NECPAL-ICO-CCOMS©3.1 tool includes the Surprise Question (SQ) and other parameters based on a multidimensional, situational assessment of needs and clinical indicators. The SQ is a subjective assessment of a patient’s life expectancy by a professional who is familiar with the patient and his/her progress (“Would you be surprised if this patient died within the next year?”). A positive response (i.e., “I would not be surprised”) acts as a trigger for ‘the palliative gaze’, i.e., for assessing the other parameters in the instrument, as it is used as a screening tool in the identification of individuals with PCNs [11].

The utility of the predictive value of the SQ has been questioned, even as part of the NECPAL instrument itself. The prognostic accuracy of the SQ and NECPAL was found to be 52.9% and 55.2%, respectively, with predictive validity being slightly higher for NECPAL [15]. Therefore, these are valuable tools for identifying patients with limited life expectancy who may require palliative care. However, the use of the NECPAL tool without the SQ may be more suitable for assessing these needs among older individuals with complex chronicity progressing to an advanced state, including many nursing home residents for whom a 12-month survival may be too short and unrealistic.

Nevertheless, given their complexity and the changing nature of the course of their illness, an objective indicator or indicators are needed to assess when to identify elderly individuals in need of palliative care. There is little scientific knowledge regarding these indicators. For this reason, we have kept the SQ in our study, as well as including the following instruments in line with the contemporary palliative care paradigm [16]: The Frail-VIG Index [17], the Case Complexity Index (CaCI) [18], and the Diagnostic Instrument of Complexity in Palliative Care (IDCPal). In addition, the following objective prognostic instruments were included in the proposed assessment to support the SQ in the NECPAL: The Palliative Prognostic Index (PPI) [19], the Palliative Performance Status (PPS) [20], and the PROFUND index [21].

Although current trends in palliative care emphasize the importance of identifying PCNs among patients, prognosis continues to be a very important factor in the referral of patients to specialized palliative care services, mainly due to professionals’ lack of resources and training in end-of-life care [15,22,23].

Based on the above, we established the following hypotheses: (1) Older residents with complex chronicity progressing to advanced chronicity in the nursing homes under study will have PCNs according to the NECPAL, regardless of whether the SQ was positive or negative, and these needs may be different and vary in number between residents with a positive SQ (SQ+) and residents with a negative SQ (SQ−); (2) characteristics such as frailty, clinical complexity, palliative care complexity, and prognostic values as measured using the PPS, PPI, and PROFUND tools will differ significantly between SQ+ residents and SQ− residents; (3) associations between variables will differ significantly between SQ+ residents and SQ− residents, as well as within the sample as a whole.

The study objectives were as follows:(1)To identify PCNs and limited life expectancy among the nursing home residents assessed using the NECPAL-ICO-CCOMS©3.1 tool, including the SQ, as well as their levels of frailty, clinical complexity, palliative care complexity, and prognostic values using specific tools.(2)To compare the values obtained by the group of SQ+ residents and the group of SQ− residents, as well as any associations between the variables analyzed in each group and in the total sample.(3)To establish the most suitable indicators based on the influence and weight of the variables analyzed.

## 2. Materials and Methods

### 2.1. Design

An observational, descriptive, cross-sectional study was carried out in 7 nursing homes in Andalusia, Spain.

### 2.2. Study Population and Sample Selection

The study population consisted of older individuals with complex chronicity residing in nursing homes. Their complex chronicity status was determined based on their chronic condition or conditions, as well as on health and/or social complexity criteria, such as recurrent hospital admissions and/or lack of family support, among others [24].

An intentional sampling method was used, which required the selection of nursing homes and, subsequently, the selection of residents from the participating nursing homes.

Nursing homes were selected based on the following criteria: Having more than 60 beds and having a multidisciplinary team with professionals who had worked at the nursing home for more than 6 months and wished to participate in the study.

The selection of residents took place as part of the NU-HELP project, in which the following inclusion criteria were used:-Residents staying at the nursing home indefinitely.-Based on the literature consulted, the prevalence of conditions in the nursing homes was distributed using the following proportions as a reference, but reflecting the real population of residents in each participating nursing home:-20% cancer patients.-40% residents with diagnosed dementia.-40% residents with specific organ failure (heart failure, renal failure, liver failure, or COPD).-Consent to participate in the study.

Residents included in family respite care programs and residents who had an established palliative care plan or were being cared for by a specific palliative care team were excluded.

Nursing homes provided their lists of users, these were randomized and 149 patients who met the inclusion criteria were selected.

### 2.3. Variables and Instruments

A structured assessment notebook was prepared, including sociodemographic variables (age and sex), clinical variables, such as existing conditions and comorbidity (using the Charlson Comorbidity Index, or CCI [25]), and the following assessment instruments:-The NECPAL-ICO-CCOMS©3.1 [11] identifies PCNs in different clinical settings. It performs non-dichotomous, indicative, multi-factor, quanti-qualitative assessments, including a subjective assessment of patient prognosis using the SQ (“Would you be surprised if this patient died within the next year?”), along with 9 sets of physical and psychosocial needs. Patients with a positive answer to the SQ (“I would not be surprised if the patient died within the next year”) and any of the aforementioned needs are considered to be ACPs. In addition, the SQ alone has been shown to be predictive of 12-month survival [15].-The Frail-VIG Index (VIG is the Spanish abbreviation for CGA (Comprehensive Geriatric Assessment)) [17] assesses 25 types of deficits through simple questions, including functional, nutritional, cognitive, emotional, and social aspects, geriatric syndromes, symptoms, and specific conditions. Based on the score obtained, the degree of frailty is classified as follows: No frailty/pre-frailty (<0.20 points), initial frailty (0.20–0.35 points), intermediate frailty (0.36–0.50), and advanced frailty (>0.50 points).-The Case Complexity Index (CaCI) [18] differentiates between clinical management complexity and community management complexity through 14 items, which are assigned a weighting and assessed in terms of severity, polypathology, skin, admissions, visits to a hospital emergency department, polypharmacy, technology required, technical support, dependency, falls, caregivers, environment, disabling individual factors, and socio-familial issues. A patient scoring ≥100 points is considered to be a complex patient, provided that the overall score for the clinical management complexity items (severity, polypharmacy, skin, admissions, and visits to a hospital emergency department) is ≥50 points.-The Diagnostic Instrument of Complexity in Palliative Care (IDCPal) [26] is a tool for diagnosing complexity in patients with advanced and end-stage disease. It contains 36 items grouped into three dimensions (relating to the patient, their relatives, and the healthcare organization), which are classified into two levels: Complexity and high complexity elements. This makes it possible to identify whether the situation is: Non-complex (there are no complexity elements present), complex (there is at least one complexity element present), or highly complex (there is at least one high complexity element present). Depending on the degree of complexity, the tool guides the practitioner in selecting the appropriate resources for the patient.-The Palliative Prognostic Index (PPI), recalibrated for advanced medical conditions [19]. This prognostic instrument assesses poor prognostic factors through 5 dimensions: Functional impairment in performing basic activities (measured on the PPI through the Palliative Performance Status (PPS) [20]), delirium, severe dyspnoea, and low oral intake. This instrument provides an approximate survival estimate based on the score obtained. In this study, we took into account the positive predictive value of the recalibrated version for advanced disease with the cut-off point at six months. The PPS included in the PPI is in itself a six-month prognostic tool. This instrument assigns a performance status percentage to each patient ranging from 0% (deceased patient) to 100% (normal, no signs of disease). The instrument provides guidance on care delivery depending on the percentage obtained, differentiating between: No need for special care (80–100%), need for some type of care (50–70%), and need for care equivalent to hospitalization or institutionalization (0–40%).-The PROFUND Index [21] contains five sections including demographic characteristics, clinical variables, analytical parameters, cognitive/functional/social variables, and healthcare-related variables. It makes it possible to predict risk of death within one year according to the score obtained: 0–2 points (16% likelihood), 3–7 points (22% likelihood), >7 points (34% likelihood).

### 2.4. Procedure

Once the nursing homes and the professionals and residents who agreed to participate in the study had been selected, professionals were specifically trained to use the tools. Data collection took place between March 2019 and February 2020. The research team remained in touch with the professionals by telephone and visited on a monthly basis to conduct follow-ups and resolve any data collection issues at the nursing homes.

### 2.5. Data Analysis

A descriptive analysis of the participants’ main characteristics was performed. Quantitative variables were described using means and standard deviations, and categorical variables were described using absolute frequencies and percentages. Quantitative data were assessed for normality using the Kolmogorov–Smirnov test, revealing a non-normal distribution. Therefore, the analysis was performed using non-parametric tests. Chi-squared tests were used to determine the independence of categorical variables. The Mann–Whitney U-test and Spearman’s correlation coefficient were used to compare and confirm the degree of association and its direction between the two independent samples (SQ+ patients and SQ− patients). The odds ratios and 95% confidence intervals for variables with statistically significant differences in both groups were also calculated. Statistical analyses were performed using IBM’s SPSS© v.25 software (IBM Corporation, Armonk, NY, USA). The statistical significance threshold for all tests was set at 0.05.

### 2.6. Ethical Oversight

All participants or their proxies (in the case of patients with cognitive impairment) gave their informed consent. The study was approved by the Research Ethics Committee for the Andalusian Public Health System in Granada (reference number: AP-0105-2016). Patient data were anonymized in compliance with the Regulation (EU) 2016/679 of the European Parliament and of the Council of 27 April 2016 (GDPR) and Spanish Organic Law 3/2018, of the 5th of December, on Personal Data Protection and Guarantee of Digital Rights.

## 3. Results

A total of 149 nursing home residents were selected and assessed, 67.1% of whom were female. The mean age was 84.47 years (±9.126 years). The most prevalent condition among residents was dementia (45.6%) and the second most prevalent was chronic heart disease (CHD), which was present in 38.3% of residents. Some patients had several coexisting conditions (Table 1).

Residents had a mean score of 2.58 (±1.98) on the CCI, suggesting a low level of comorbidity. The mean CaCI score was 103.99 points, with 65.8% of patients being complex cases. Frailty, as measured by the Frail-VIG Index, stood at a mean of 0.28 points (±0.073), with 61.1% of residents having initial frailty and 19.5% having intermediate frailty. A total of 53% of patients were SQ+, along with several other NECPAL ICO-CCOMS© parameters, allowing them to be classified as advanced complex chronic patients. As measured by the IDCPal, 43% of residents displayed palliative care complexity and 22.8% high palliative care complexity.

The mean palliative performance of the sample, as measured by the PPS, was 64.43%, while the PPI showed a mean score for residents of 2.48 (±2.57). Translated to the scores provided by the PPI, this suggests that 42% of patients with similar characteristics could die within six months. However, based on the mean score obtained by the sample on the PROFUND index, between 45% and 50% of patients would be likely to die within a year.

Table 1 shows other sociodemographic and clinical data for the sample.

The most prevalent need among all residents as listed in the NECPAL ICO-CCOMS© tool (Table 2) was the presence of Distress and/or Severe Adaptive Disorder (97.3%), followed by Specific Indicators, which were present in 73.8% of residents. The most frequent need in both the PSQ+ and PSQ− groups was Distress and/or Severe Adaptive Disorder, coinciding with the total group. The second most frequent need in the PSQ+ group was Specific Indicators, while in the PSQ− it was Functional Decline, with a prevalence of 38.9% in that group.

Statistically significant differences in the presence of Nutritional Decline (*p* = 0.017) (OR = 5.500; 95% CI = (1.175; 25.750)), Cognitive Decline (*p* = 0.032) (OR = 2.042; 95% CI = (1.058; 3.943)), Severe Dependence (*p* = 0.006) (OR = 3.138; 95% CI = (1.347; 7.307)), and Specific Indicators (*p* < 0.001) (OR = 4.121; 95% CI = (1.858; 9.140)) between the PSQ+ and PSQ− groups were found. These needs were more prevalent in the PSQ+ group, while Functional Decline (*p* < 0.001) (OR = 0.272; 95% CI = (0.129; 0.575)) was more prevalent in the PSQ− group. These variables were significantly associated with a positive SQ.

In relation to Case Complexity (Table 3), no statistically significant differences (*p* = 0.297) between the PSQ+ and PSQ− groups were found. However, the level of Frailty as measured by the Frail-VIG Index showed statistically significant differences in intermediate frailty between these groups (*p* = 0.005) (OR = 3.474; 95% CI = (1.380; 8.742)). Regarding Palliative Care Complexity, statistically significant differences in the presence of Complexity were identified between the two groups (*p* = 0.032) (OR = 1.966; 95% CI = (1.014; 3.811)). These variables were also found to have a significant influence on the likelihood of receiving a positive SQ.

The Spearman’s correlation coefficients showing significant relationships in the total sample and in the PSQ+ and PSQ− groups produced the following results.

As shown in Table 4, in the total sample, positive correlations were identified between the number of Positives on the NECPAL and the Frail-VIG score. Negative correlations between the PPS score and the CaCI, Frail-VIG, and PROFUND scores and between the number of Positives on the NECPAL and Positives on the IDCPal were also observed. A negative correlation between the PPS score and the PPI score (R = −0.631, *p* = 0.01) was found, suggesting that lower PPS scores are associated with higher PPI scores.

In the group of SQ+ residents (Table 5), significant negative correlations between the PPS score, the CaCI score, the number of positives on the IDCPal, the PROFUND score, and the PPI score were found, the latter being negatively correlated (R = −0.587, *p* = 0.01).

In the group of SQ− residents (Table 6), we found a positive correlation between the Frail-VIG score and the PPI score (R = 0.578, *p* = 0.05). Similarly, the number of Positives on the NECPAL was positively correlated with the number of Positives on the IDCPal (R = 0.514, *p* = 0.01). As for the negative correlations in this group, we found that the variable Age was negatively correlated with the number of Positives on the IDCPal (i.e., the older the individual, the fewer positives they scored on the IDCPal). The PPS score was also negatively correlated with the Frail-VIG score, the number of Positives in NECPAL, the number of Positives in IDCPal, and the PROFUND score. The PPS score was negatively correlated with the PPI score (R = −0.570, *p* = 0.01), suggesting that lower PPS scores in this group indicate higher PPI scores and vice versa.

## 4. Discussion

This study assessed the PCNs, level of frailty, case complexity, palliative care complexity, and prognostic life expectancy values of complex chronic residents, regardless of whether the SQ included in the NECPAL tool was positive or negative, and aimed to identify the most appropriate indicators for assessing PCNs associated with progression to advanced chronicity among this population.

In the NECPAL tool [27], the SQ serves as a screening measure. A positive response to the SQ is required to implement the tool and assess the other indicators. As stated in the tool itself, a patient with a negative SQ is not NECPAL, i.e., is not eligible to be assessed for PCNs. However, our results showed that the SQ− group had PCNs as assessed with the NECPAL instrument, obtaining a number of positives on all items of the tool, albeit mostly at a lower percentage than the SQ+ group. Complex palliative care and highly complex palliative care were also present in the SQ− group, the latter even in a slightly higher proportion than the SQ+ group, confirming their status as residents with PCNs.

A recent study [15] assessed the predictive value of the SQ and the NECPAL tool at 12 and 24 months among SQ+ and NECPAL+ patients, finding that their predictive validity was significant at 24 months and slightly higher for the NECPAL tool than for the SQ. Similarly, another study [28] reported that around 60% of admitted patients who were assessed with this tool died within two years, more than 25% of whom were non-NECPAL (SQ−). Therefore, it is fairly safe to say that making assessment of PCNs conditional on one-year survival projections (SQ+) excludes a substantial proportion of patients or residents with PCNs and a longer life expectancy (SQ−).

If we consider the illness trajectories accompanying PCNs under the new paradigm in palliative care, most of the residents in the sample have advanced chronic conditions with organ dysfunctions, in some cases combined with dementia and other common commodities (e.g., diabetes, arterial hypertension, cancer, etc.). The progression of these conditions will cause residents to experience attacks in the final years of their lives, which could lead to unexpected death without their PCNs being properly identified and met [5,8]. Studies such as Blay et al. [16] show that a shift in community palliative epidemiology is taking place, with an increased focus on ageing. Residents’ illness trajectories could support the need for early assessment of PCNs, irrespective of the prognosis provided by the SQ. Our data suggest that the presence of nutritional and cognitive decline, as well as severe dependency and specific NECPAL indicators, had a significant influence on the likelihood of receiving a positive SQ. Although these variables were also present in SQ− residents, the differences between the two groups were significant. The presence of these variables when assessing PCNs using the NECPAL tool could help to identify ACPs among nursing home residents, who, as mentioned earlier, have an uncertain prognosis depending on their changing situation.

Therefore, these data suggest that, in a population whose complex chronicity rapidly and uncertainly progresses to advanced complex chronicity, the use of the SQ (a 12-month subjective prognosis) as a prerequisite for conducting a full assessment to identify ACPs [27] using the NECPAL CCOMS-ICO© tool may not be the most adequate approach. For this reason, we consider it necessary to establish a criterion to support or replace the SQ in populations such as this.

Level of frailty was one of the characteristics showing statistically significant differences between the SQ+ and SQ− groups in our sample. Few studies have assessed the level of frailty in a similar setting. Vivanco et al. [29] reported higher figures than ours; this is likely to be due to the fact that they only assessed residents aged 85 and over, with a mean age of 90.9 (±4.2) years compared to 84.47 (±9.126) in our study.

Frailty is a concept that is closely linked to advanced chronicity and an essential aspect in geriatric assessment [30]. It accompanies the latter in most of the studies on nursing home populations that have been consulted [16,17,31,32,33]. Both nationally and internationally, frailty has traditionally been assessed using criteria such as the Fried frailty phenotype or the Rockwood Frailty Index [34,35]. Authors such as Amblàs-Novellas et al. have recently proposed the use of frailty indices with geriatric patients, specifically the Frail-VIG instrument [17], as a more relevant assessment for complex clinical situations. This index, based on a multidimensional assessment and with a good discriminating power of the level of frailty, may be an objective, determining factor in assessing the PCNs of nursing home residents.

Our results support this proposal, as we found that intermediate levels of frailty were associated with the SQ. Furthermore, the correlation analysis of both the SQ+ and SQ− groups confirmed the relationships between the Frail-VIG score and case complexity (CaCI), positive NECPAL scores, and the PPI prognostic tool. This suggests that the level of frailty (i.e., the level of vulnerability due to accumulated deficits, or remaining health) [17,36] and, more specifically, the intermediate level of frailty, may be an alternative to a positive SQ in measuring advanced disease progression (an individual’s life stage) [17] and therefore in determining when to assess PCNs. Likewise, the prognostic ability of the frailty index may be helpful in making early, objective palliative care decisions.

Regarding case complexity (CaCI), no statistically significant differences between the SQ+ and SQ− groups were found. Therefore, the SQ had no discriminating power among nursing home residents, as they exhibited case complexity regardless of whether they were SQ+ or SQ−. Complexity is one of the pillars of palliative care, and understanding it allows teams of professionals to deliver interventions in accordance with the patient’s actual condition [37,38]. Both the CaCI and the IDCPal [26] used in this study provided the professionals in the participating health and social care teams with valuable information on their residents and allowed them to work directly on aspects where complexity was present. Further research is required to confirm these results, although studies such as Esteban-Pérez et al. [37] and Salvador-Comino et al. [39] point in a similar direction.

There were also no significant differences in Palliative Care Complexity. However, interestingly, a high proportion of SQ− residents scored higher on palliative care complexity items than SQ+ residents. This may give the impression that the real complexity experienced by nursing home residents is palliative in nature. Residents who would not have been considered ACPs according to the traditional NECPAL criteria (SQ−) have very similar results to SQ+ residents or ever poorer results than SQ+ residents in highly complex palliative cases. No studies comparing palliative care complexity among nursing home residents were found, with most studies assessing palliative care complexity among palliative care patients [39,40].

Further in-depth studies are needed to determine the exact reasons for this apparently contradictory situation. However, a more detailed analysis of the scored items suggests that it could be due to a greater presence of high complexity criteria within the psycho-emotional situation of SQ− residents and their dependency on family and the people around them. It could also be due to the subjective nature of the assessments made by the professionals or to the data on which they based their assessments. For that reason, it is necessary to reinforce professionals training in the correct use of tools for the assessment and detection of palliative needs [13].

Prognostic value is a common feature of most of the instruments used in the type of assessment proposed in this study. This is because it is currently an essential aspect of practice, guiding the use of resources and the provision of care in a diligent, prioritised, and situation-appropriate manner [22,23,41,42]. Our study confirms that residents who show greater frailty [43], greater comorbidity [41], and lower performance as per the PPS [22] have a more limited life expectancy and higher scores on the prognostic instruments used (the PPI and PROFUND) [19,21,22,41]. We also observed that these variables were positively correlated with one another in most of the cases in our sample. Future follow-up studies are needed to evaluate the prognostic utility of the instruments in our sample, including the NECPAL tool and the SQ. Nevertheless, their utility is expected to be confirmed, as was the case of studies conducted in nursing homes and other settings, such as Rice et al. [43] and Gómez-Batiste et al. [15].

From a clinical utility perspective, the use of a prognostic index such as the PPI or the PROFUND Index [19,21,22] in nursing homes could support the assessment made by the SQ within the NECPAL instrument, allowing the professional conducting the assessment to prioritize and make a time estimate of the care interventions and/or referrals to specific resources required by residents. However, the priority in determining when to assess residents’ PCNs is not to replace the SQ with another prognostic tool, but rather for the prognosis to serve as an indicator of the patient’s status in order to adjust care goals and optimize the use of resources [44,45,46]. In this sense, the prognostic ability of the level of frailty can be helpful, as noted above.

This study has a number of limitations that should be taken into account. Most of the instruments used for our assessment were developed and used in Spain, so there may be limitations when comparing the results with studies carried out in other countries. In addition, some of these instruments have not previously been used to assess nursing home residents, which may produce varying results due to the complexity of this population’s background, needs, and care. It is also important to note that a follow-up of the residents in the sample is needed to confirm some of the results obtained and the utility of conducting the proposed assessment in nursing homes.

## 5. Conclusions

Elderly individuals with complex chronicity progressing to advanced chronicity in the nursing homes under study exhibited PCNs under the NECPAL CCOMS-ICO© tool, regardless of whether the response to the SQ was positive or negative.

For this reason, we argue that ascertaining an intermediate level of frailty among residents using the Frail-VIG index could replace the SQ in the NECPAL CCOMS-ICO© tool when assessing PCNs and, consequently, when identifying ACPs among elderly people in nursing homes. In addition, we suggest that the detection of different types of decline, severe dependency, or specific indicators using the NECPAL tool is also helpful in identifying ACPs among residents.

Finally, we believe that it is necessary to adopt assessments such as the one proposed in our study as standard practice in nursing homes, combining objective tools for the assessment of PCNs; characteristics such as the level of frailty, case complexity, and complexity of palliative care helping professionals to understand residents’ real needs and circumstances; and objective prognostic instruments allowing the use of resources and provision of care to be prioritized.

## Figures and Tables

**Table 1 ijerph-18-03171-t001:** Sociodemographic and clinical characteristics of patients.

Variables	Total Sample (*n* = 149) (M (SD)/*n* (%))		PSQ+ (*n* = 79) (M (SD)/*n* (%))		PSQ− (*n* = 70) (M (SD)/*n* (%))		*p*
Age	84.47	(±9.126)	85.3	(±9.482)	83.5	(±8.672)	0.050 ^a^
Female	100	(67.1)	58	(73.4)	42	(60)	0.082 ^b^
Male	49	(32.9)	21	(26.6)	28	(40)	
Coexisting conditions							
Cancer	24	(16.1)	14	(17.7)	10	(14.3)	0.569 ^b^
CPD	34	(22.8)	16	(20.3)	18	(25.7)	0.428 ^b^
CHD	57	(38.3)	38	(48.1)	19	(27.1)	0.009 ^b^
CND	12	(8.1)	7	(8.9)	5	(7.1)	0.701 ^b^
CLD	1	(0.7)	1	(1.3)	0	(0)	0.345 ^b^
CRD	19	(12.8)	11	(13.9)	8	(11.4)	0.649 ^b^
Dementia	68	(45.6)	39	(49.4)	29	(41.4)	0.332 ^b^
Patient status							
CCI	2.58	(±1.98)	2.94	(±2.366)	2.17	(±1.351)	0.043 ^a^
CaCI	103.99	(±20.17)	106.08	(±20.36)	101.64	(±19.849)	0.205 ^a^
Frail-VIG	0.28	(±0.073)	0.303	(±0.071)	0.257	(±0.069)	<0.001 ^a^
Positives IDC-Pal	1.31	(±1.537)	1.32	(±1.524)	1.30	(±1.563)	0.565 ^a^
PPS	64.43	(±18.431)	57.59	(±16.189)	72.14	(±17.847)	<0.001 ^b^
PPI	2.48	(±2.572)	3.16	(±2.731)	1.72	(±2.155)	0.001 ^b^
PROFUND	8.81	(±4.153)	10.03	(±3.883)	7.44	(±4.046)	<0.001 ^b^

PSQ+, patients with a positive surprise question; PSQ−, patients with a negative surprise question; CPD, chronic pulmonary disease; CHD, chronic heart disease; CND, chronic neurological disease; CLD, chronic liver disease; CRD, chronic renal disease; CCI, Charlson Comorbidity Index; CaCI, Case Complexity Index; M, mean; SD, standard deviation; ^a^: Mann-Whitney; ^b^: Chi-squared.

**Table 2 ijerph-18-03171-t002:** Comparison and association of the presence of NECPAL ICO-CCOMS©3.1 items between groups.

No. Item	NECPAL ICO-CCOMS©3.1 Items	Total (*n* = 149) (%)	PSQ+ (*n* = 79) (%)	PSQ− (*n* = 70) (%)	*p* ^b^	OR (95% CI)
1	Positive Surprise Question	53	100	0	-	
2	Demand	4	3.4	0.7	0.129	
	Need identified by healthcare professionals in the team	18.1	12.1	6	0.116	
3	Nutritional Decline	8.7	7.4	1.3	0.017	5.500 (1.175; 25.750)
	Functional Decline	67.8	28.9	38.9	<0.001	0.272 (0.129; 0.575)
	Cognitive Decline	56.4	34.2	22.1	0.032	2.042 (1.058; 3.943)
4	Severe Dependence	22.8	16.8	6	0.006	3.138 (1.347; 7.307)
5	Geriatric Syndromes	40.9	23.5	17.4	0.375	
6	Persistent Symptoms	47	24.8	22.1	0.970	
7	Distress and/or Severe Adaptive Disorder	97.3	50.3	47	0.056	
	Severe Social Vulnerability	2	0.7	1.3	0.490	
8	Multimorbidity	4.7	2	2.7	0.581	
9	Use of Resources	54.4	29.5	24.8	0.728	
10	Specific Indicators	73.8	45.6	28.2	<0.001	4.121 (1.858; 9.140)

^b^ Chi-squared; OR (95% CI), Odds Ratio (95% Confidence Interval).

**Table 3 ijerph-18-03171-t003:** Comparison and associations between the Case Complexity Index (CaCI), Frail-VIG, and Diagnostic Instrument of Complexity in Palliative Care (IDCPal) results for the total group and for residents with a positive Surprise Question (PSQ+) and a negative Surprise Question (PSQ−).

Instrument Results	Total (*n* = 149) (%)	PSQ+ (*n* = 79) (%)	PSQ− (*n* = 70) (%)	*p* ^b^	OR (95% CI)
Complex (CaCI)	65.8	36.2	29.5	0.297	
Initial Frailty (Frail-VIG)	61.1	32.2	28.9	0.534	
Intermediate Frailty (Frail-VIG)	19.5	14.8	4.7	0.005	3.474 (1.380; 8.742)
Complex (IDCPal)	43	26.8	16.1	0.032	1.966 (1.014; 3.811)
Highly Complex (IDCPal)	22.8	10.1	12.8	0.162	

^b^ Chi-squared; OR (95% CI), Odds Ratio (95% Confidence Interval).

**Table 4 ijerph-18-03171-t004:** Spearman correlation matrix for the total sample.

Total Sample (*n* = 149)	Age	CaCI	Frail-VIG	Positives (NECPAL)	Positives (IDCPal)	PROFUND	PPS	PPI
Age	1							
CaCI	0.132	1						
Frail-VIG	−0.068	0.395 **	1					
Positives (NECPAL)	0.104	0.133	0.405 **	1				
Positives (IDCPal)	−0.130	−0.002	0.168 *	0.375 **	1			
PROFUND	0.060	0.198 *	0.342 **	0.148	0.062	1		
PPS	−0.121	−0.273 **	−0.330 **	−0.374 **	−0.339 **	−0.465 **	1	
PPI	0.130	0.240 **	0.478 **	0.444 **	0.264 **	0.481 **	−0.631 **	1

* Significance at 0.05 level; ** Significance at 0.01 level.

**Table 5 ijerph-18-03171-t005:** Spearman correlation matrix for the PSQ+ group.

PSQ+ (*n* = 79)	Age	CaCI	Frail-VIG	Positives (NECPAL)	Positives (IDCPal)	PROFUND	PPS	PPI
Age	1							
CaCI	0.129	1						
Frail-VIG	−0.117	0.485 **	1					
Positives (NECPAL)	0.147	0.315 **	0.296 **	1				
Positives (IDCPal)	0.020	0.104	−0.037	0.301 **	1			
PROFUND	−0.143	0.205	0.174	−0.077	−0.023	1		
PPS	−0.021	−0.280 *	−0.089	−0.163	−0.247 *	−0.356 **	1	
PPI	0.075	0.343 **	0.263 *	0.382 **	0.126	0.421 **	−0.587 **	1

* Significance at 0.05 level; ** Significance at 0.01 level.

**Table 6 ijerph-18-03171-t006:** Spearman correlation matrix for the PSQ− group.

PSQ− (*n* = 70)	Age	CaCI	Frail-VIG	Positives (NECPAL)	Positives (IDCPal)	PROFUND	PPS	PPI
Age	1							
CaCI	0.106	1						
Frail-VIG	−0.138	0.257 *	1					
Positives (NECPAL)	−0.155	−0.119	0.317 **	1				
Positives (IDCPal)	−0.334 **	−0.125	0.348 **	0.514 **	1			
PROFUND	0.204	0.111	0.375 **	0.042	0.086	1		
PPS	−0.045	−0.151	−0.274 *	−0.250 *	−0.453 **	−0.347 **	1	
PPI	0.111	0.078	0.578 *	0.341 **	0.370 **	0.449 **	−0.570 **	1

* Significance at 0.05 level; ** Significance at 0.01 level.

## Data Availability

The data presented in this study are available on reasonable request from the corresponding author. The data are not publicly available due to privacy restrictions.

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
