# Peer review of "A New Approach to the Identification of Palliative Care Needs and Advanced Chronic Patients among Nursing Home Residents"

_ijerph, 2021, doi:10.3390/ijerph18063171_

Round 1
Reviewer 1 Report
Dear Reserchers,
I have only two issues (which should be improved):
1)Abstract can be improved (the research "conducted in 149 nursing homes Complex Chronic residents", but your research group included also professionals and this information is no in abstract;
2)The table 1 presented (among other)sociodemographic data for the sample should be improved. As you included in the table female, in my opinion you should also include male. As the result the difference between women and men will be more visible
Author Response
Please, see the attachment.
Thank you,
Ana A. Esteban

Reviewer 2 Report
General comments
The study addresses an important clinical issue.Staff working in residential aged care facilities (RACF) must be able to recognise deterioration. Some discussion about staff competence and necessary training to do so could be included.
Life expectancy in RACF is generally about 3 years.
Indicate when the initial assessment and any follow-up assessments should be undertaken..
Note palliative care is not reserved for people who are dying or the last year of life. Better outcomes are achieved in it is introduced early.
20-27% of people living in RACFs have diabetes and usually common commodities. This could be included in the discussion of common commodities e.g. cancer. Diabetes is an under-reported underlying cause of death but is in the top 3 leading causes of death.
Methods
Indicated how residents were selected.
Amend to inclusive language e.g. people/person with chronic conditions, including in the title.
Author Response
Please, see attachment.
Thank you,
Ana A. Esteban
